# The Effects of Local Treatment of PTH(1-34) and Whitlockite and Hydroxyapatite Graft to the Calvarial Defect in a Rat Osteoporosis Model

**DOI:** 10.3390/biomedicines12040820

**Published:** 2024-04-08

**Authors:** Jiwoon Jeong, Jung Hee Shim, Chan Yeong Heo

**Affiliations:** 1OSFIRM R&D Center, H&BIO Co., Ltd., Seongnam-si 13605, Republic of Korea; jwjeong@osfirm.co.kr; 2Interdisciplinary Program in Bioengineering, Seoul National University, Seoul 08826, Republic of Korea; 3Department of Plastic and Reconstructive Surgery, Seoul National University Bundang Hospital, Seongnam-si 13620, Republic of Korea; xmylife@empas.com; 4Department of Plastic and Reconstructive Surgery, College of Medicine, Seoul National University, Seoul 08826, Republic of Korea

**Keywords:** osteoporosis, bone regeneration, osteoclast, parathyroid hormone, bone graft, whitlockite, hydroxyapatite

## Abstract

With the aging population, there is a rising incidence of senile diseases, notably osteoporosis, marked by fractures, prolonged recovery, and elevated mortality rates, underscoring the urgency for effective treatments. In this study, we applied the method of absorbing parathyroid hormone (PTH), a treatment for osteoporosis, into graft materials. Two types of graft materials with different properties, whitlockite (WH) and hydroxyapatite (HAP), were used. After forming calvarial defects in osteoporotic rats, WH and HAP grafts were implanted, with PTH applied directly to the graft sites. Micro-CT analysis was employed to assess bone regeneration, while tissue sections were stained to elucidate the regeneration process and bone cell dynamics. The results showed that bone regeneration was higher in the grafts that were actively degraded by osteoclasts in the early stage of regeneration. When PTH was applied, osteoclast activity increased, leading to enhanced bone regeneration. Furthermore, the activation of osteoclasts resulted in the penetration and formation of new bone within the degraded graft, which exhibited higher osseointegration. Therefore, for osteoporotic bone defects, bone grafts that can be easily degraded by osteoclasts are more suitable. Additionally, treatment with PTH can activate osteoclasts around the bone graft in the early stages of regeneration, inducing higher bone regeneration and improving osseointegration.

## 1. Introduction

The global population’s accelerating aging trend has increased the demand for treating senile diseases. Osteoporosis, in particular, has a global prevalence rate exceeding 18% across all age groups, with a steady annual increase [1,2,3]. Additionally, the number of patients with osteoporotic fractures is also significantly increasing every year [1,2,4,5,6]. Osteoporotic fractures are difficult to treat and often result in refractures and high mortality rates. Therefore, more effective treatments for osteoporotic fractures are needed [2,4,5,6].

The treatment of osteoporosis typically involves the administration of drugs targeting mechanisms that regulate bone remodeling. Among these drugs are bisphosphonates, which inhibit bone resorption [7,8], and parathyroid hormone (PTH) drugs, which stimulate bone formation [9]. However, bisphosphonates have limitations that restrict their long-term use, such as the potential to induce osteonecrosis in certain patients. Consequently, there is a growing demand for parathyroid hormone drugs [10,11].

PTH influences blood calcium concentration and bone metabolism. It acts on bone cells to enhance cell activity and osteogenic differentiation, primarily by stimulating the Wnt signaling pathway. This activation of bone cells promotes the normalization of bone remodeling and inhibits osteoporosis [12,13,14,15]. Nonetheless, rhPTH(1-34), a representative parathyroid hormone drug, requires administration via injections, which may inconvenience patients due to the need for periodic injections. Therefore, there is a need to find a more effective and convenient method of drug administration.

Synthetic bone grafts are increasingly replacing allogeneic and xenogeneic bone grafts in the treatment of bone defects due to the risk of disease infection and ethical concerns. However, their limited osteoinductive capacity restricts their use in patients with difficult bone regeneration, such as osteoporosis [16]. Therefore, there is a growing need for synthetic bone graft materials with higher efficacy. Hydroxyapatite (HAP) is a conventional synthetic bone graft material that is widely used due to its high safety in the body and numerous studies [17,18]. Recently, whitlockite (WH) has gained attention as a new synthetic bone graft material. WH has the chemical formula Ca_18_Mg_2_(HPO_4_)_2_(PO_4_)_12_ and is also found in the body [17,19]. Research on WH has been actively conducted recently due to its higher bone regeneration efficacy and faster degradation by osteoclasts in the body compared to HAP [17,20].

Therefore, the incorporation of WH, a novel bone graft material, alongside HAP, is anticipated to yield more effective bone regeneration in conditions where bone regeneration is challenging, such as osteoporotic fractures. Moreover, the concurrent administration of PTH, known to augment bone regeneration, during bone grafting is expected to synergize with the two materials, fostering effective bone regeneration.

In this study, we applied PTH directly to the defect during bone grafting to effectively treat osteoporotic bone defects. We compared the bone regeneration process under osteoporosis by applying two types of bone grafts, WH and HAP, each with distinct properties. Calvarial defects were induced in rats with osteoporosis through ovariectomy. The regeneration process was analyzed by applying different bone grafts along with PTH. This study aimed to compare the suitability of bone graft materials for osteoporotic defects and to assess the efficacy of directly applying PTH to the defect, aiming to explore more effective treatments for osteoporotic bone defects.

## 2. Materials and Methods

### 2.1. Bone Graft Preparation

Bone graft was fabricated by a polyurethane sponge replica method [21]. The coating slurry was prepared by mixing 55 wt% calcium phosphate ceramic powder and 7 wt% polyvinyl butyral (PVB, M.W. 40,000~70,000, Aladdin, Riverside, CA, USA) in ethanol solvent using a planetary mill. The calcium phosphate ceramics used were whitlockite (WH, CAS NO. 106984-49-0, H&Bio Co., Ltd., Seongnam-si, Republic of Korea) and hydroxyapatite (HAP, CAS NO. 1306-06-5, Daejung chemical, Siheung-si, Republic of Korea), respectively. Polyurethane sponges with a pore size of 50 ppi were fully immersed in the prepared slurry and spun at 800 rpm using a spin coater to evenly coat them without clumping. The sponges were dried at 60 °C for three hours. This coating process was repeated four times. Next, the bone graft was treated in a furnace at 500 °C for 2 h to remove the remaining polymer and sintered at 1000 °C for 2 h to produce the bone graft. The fabricated bone grafts were then sieved to a size of 1 mm to 600 μm and sterilized prior to use.

### 2.2. X-ray Diffraction Analysis

X-ray diffraction (XRD, MiniFlex600, Rigaku, Tokyo, Japan) measurements were performed to confirm the crystal structure of the fabricated bone graft. The measurements were taken from 10 to 70° at a rate of 1°/min. The measured data were matched with WH (9012137) and HAP (9013627) from Crystallography Open Database (COD) and confirmed that the fabricated WH and HAP bone grafts were formed from pure WH and HAP.

### 2.3. Animal Surgery and Treatment

The animal studies were performed after receiving approval from the Institutional Animal Care and Use Committee (IACUC) of Seoul National University Bundang Hospital. Seventy-five female Sprague Dawley (SD) rats weighing 220 ± 10 g were used to create an osteoporosis model through ovariectomy (OVX). The ovariectomy was performed under respiratory anesthesia with isoflurane (Ifran Liq., Hana Pharm, Seoul, Korea) by incising the outer and inner skin at the ovarian position of rats, exposing the ovary, and then resecting it. The inner and outer skin were then sutured and sanitized to complete the ovariectomy. Subsequently, bone mineral density (BMD) was monitored using dual-energy X-ray absorptiometry (DEXA, Inalyzer, Medikors, Seongnam-si, Republic of Korea) to determine the presence of osteoporosis. At week 13, a decrease in BMD of over 30% was observed, confirming the development of osteoporosis. In week 14, a calvarial defect model was performed. To create the calvarial defect, osteoporotic rats were anesthetized and a sagittal incision was made over the middle sagittal crest. An 8 mm diameter defect was then created using a trephine burr. Each group was implanted with 30 mg of the following bone grafts. There were five experimental groups: the study included a sham control group (sham) without bone grafts and groups with whitlockite grafts (WH), hydroxyapatite grafts (HAP), whitlockite grafts with PTH drug (WH + PTH), and hydroxyapatite grafts with PTH drug (HAP + PTH). After the bone graft was placed, the group receiving PTH drug was treated with 56.5 mg/kg teriparatide (parathyroid hormone fragment 1-34 human, EP reference standard, Sigma-Aldrich, Burlington, MA, USA) dissolved in 20 μL saline and thoroughly absorbed onto the graft material (Appendix A). The incisions were then sutured and sanitized to complete the surgery. At 4, 8, and 12 weeks post-surgery, five rats from each group were euthanized using CO_2_ gas. The surgical sites were then harvested and fixed in formalin solution (Formaldehyde 3.7% Solution, Biosesang, Yongin-si, Republic of Korea) for storage.

### 2.4. Micro-CT Analysis

The harvested bone samples were measured by microcomputed tomography (micro-CT, Skyscan1173, Bruker, Billerica, MA, USA) at 130 kV, 60 μA, 500 ms. The measured data were reconstructed for 3D analysis, and the volume and surface area of the bone and graft materials were calculated using software (CTAn, Bruker, Germany). Since the measured radiodensity of the remaining graft overlapped with the radiodensity of the regenerated bone and a clear separation of the remaining graft and the new bone was not possible, the value including both the volume of the new bone and the remaining graft was calculated as bone volume (BV) and the value including both the surface area of the new bone and the remaining graft was calculated as bone surface area (BS).

### 2.5. Histological and Histomorphometric Analysis

For histological analysis, the harvested tissues were decalcified using ethylenediaminetetraacetic acid (EDTA, Daejung chemical, Siheung-si, Republic of Korea) at pH 7.2 for one week. Subsequently, the decalcified tissues were embedded in paraffin and sectioned for staining. Three types of stains were used: hematoxylin and eosin (H&E), Masson’s trichrome (MT), and tartrate-resistant acid phosphatase/methyl green (TRAP). H&E staining was performed to analyze the morphology and cell behavior of the tissue formed at the defect site, and MT staining was performed to analyze the formation of new bone and collagen fibers. TRAP staining was used to analyze the distribution and activity of osteoclasts. The stained slides were examined under a light microscope at ×60 magnification. Histomorphometric analysis was calculated by measuring the area of tissue corresponding to each connective tissue (CT), new bone (NB), and remaining bone graft (BG) using the ImageJ program (https://imagej.net/).

### 2.6. Statistical Analysis

Statistical analysis was performed using one-way ANOVA test, followed by Tukey test. Statistical significance was considered based on the *p*-value: * *p* < 0.05 vs. Sham group, ** *p* < 0.01 vs. Sham group, *** *p* < 0.001 vs. Sham group, # *p* < 0.05, ## *p* < 0.01, and ### *p* < 0.001. Post hoc power analysis using G*Power software (version 3.1.9.7) showed that all statistical analyses had a power exceeding 0.8.

## 3. Results

### 3.1. Bone Graft Preparation

In this study, both WH and HAP bone grafts were manufactured using the same production process to ensure equitable comparison. The raw materials of each bone graft were mixed in a PVB/EtOH slurry, coated on a PU sponge to form a structure, and sintered at 1000 °C to produce the bone graft. Each bone graft material was crushed to a size of 1000~600 μm and utilized as bone graft material (Appendix A). XRD measurements of the fabricated bone grafts indicated that the XRD peaks of WH and HAP were well defined, confirming the formation of the bone grafts without impurities (Figure 1).

### 3.2. Animal Model Preparation

To analyze the behavior of the bone graft and the efficacy of PTH drug under the osteoporosis model, the rat OVX model was utilized to induce osteoporosis. OVX was performed on female SD rats weighing 220 ± 10 g, and the progression of osteoporosis was assessed by analyzing BMD through DEXA measurements over a period of time. After 13 weeks, a BMD decrease of more than 30% was confirmed, indicating the formation of osteoporosis, and a calvarial defect model was created at week 14 to apply the experiment. Each bone graft was implanted with 30 mg in an 8 mm defect, and the PTH treatment group was treated with 56.5 mg/kg of PTH resorbed into the bone graft.

### 3.3. Micro-CT Analysis

To analyze the regenerated bone defect following grafting, each animal was sacrificed and tissue was harvested at weeks 4, 8, and 12. BV/TV was assessed by quantifying the volume of new bone and remaining graft within the defect using micro-CT analysis (Figure 2b). This analysis determined the proportion of volume occupied by new bone and remaining graft within the defect volume. Across all experimental groups, no statistical differences were observed between groups regardless of the healing week. At the initial stage of regeneration (week 4), BV/TV values did not significantly differ between WH and HAP groups, with a slight decrease in BV/TV noted in the WH + PTH and HAP + PTH groups treated with PTH. The lowest BV/TV was observed in the WH + PTH group. At week 8, all groups showed increased BV/TV values from week 4, with WH showing higher BV/TV values than HAP and WH + PTH showing BV/TV values above WH, which was different from the lowest values at week 4. HAP + PTH had the lowest BV/TV values. At week 12, WH had slightly higher levels of BV/TV than HAP, WH + PTH had the highest BV/TV, and HAP + PTH had values higher than HAP and lower than WH + PTH.

BS/TV and BS/BV were analyzed by analyzing the surface occupied by new bone and remaining graft. First, the surface of the bone formed in the volume of the defect was analyzed to determine whether the formation of new bone occurred only in a specific area or in multiple areas (Figure 2c). All groups showed similar values at week 4 with no significant differences. At week 8, WH showed increased BS/TV values compared to week 4 and WH + PTH showed the highest increase, with higher BS/TV values than WH. HAP and HAP + PTH showed BS/TV that were not significantly different from week 4. At week 12, WH showed an increase in BS/TV from week 8 and WH + PTH showed the highest BS/TV value, higher than WH. HAP showed no significant difference in BS/TV from week 8, and HAP + PTH showed a slight increase in BS/TV from week 8. Subsequently, BS/BV was analyzed to assess the density of the newly formed bone, investigating whether it exhibited high density or formed a structure with a larger surface area (Figure 2d). All experimental groups showed no statistical difference between groups regardless of the duration of regeneration (Figure 2d). The WH, WH + PTH, and HAP groups all maintained similar values at 4, 8, and 12 weeks, with WH and WH + PTH showing higher BS/BV than HAP. HAP + PTH showed a gradual decrease in BS/BV at weeks 4, 8, and 12.

### 3.4. Histological Analysis

The defect sections of each tissue were stained for histologic analysis. MT staining at week 4 showed the most bone formation around the graft in WH and some new bone formation around the graft in HAP (Figure 3a). Additionally, in the WH group, new bone penetration and formation within the graft were observed. However, both the WH + PTH and HAP + PTH groups exhibited minimal new bone formation and bone fibers.

MT staining at week 12 revealed thickening of the new bone surrounding the graft in the WH group, with the new bone penetrating the surface of the graft and replacing it with new bone (Figure 3d). In the WH + PTH group, the highest amount of new bone formation was observed, with new bone penetrating the surface of the graft and replacing it with new bone. This phenomenon appeared to be more pronounced in the WH + PTH group compared to the WH group alone. In the HAP group, new bone formation was observed adhering to the surface of the graft with a distinct boundary between the new bone and the surface of the graft. Similarly, in the HAP + PTH group, new bone formation adhered to the surface of the graft with a clear boundary, and some of the new bone penetrated into the graft.

TRAP staining at week 4 revealed high TRAP expression in the WH group (Figure 3c). High TRAP expression was observed in multinucleated cells surrounding the graft, indicating activation of numerous osteoclasts around the graft, with some osteoclasts penetrating into the graft. In the WH + PTH group, TRAP expression was even higher compared to the WH group. A substantial number of activated osteoclasts were observed on the surface of the graft, with many osteoclasts penetrating into the graft. HAP exhibited mild TRAP reactivity on the surface of the graft. In the HAP + PTH group, some osteoclasts were observed forming on the graft’s surface, with weak TRAP activation.

At week 12, the WH group exhibited the formation of some osteoclasts on the surface of the graft with weak TRAP activity (Figure 3f). In the WH + PTH group, a small number of osteoclasts formed on the graft’s surface and displayed very weak TRAP activity. In the HAP group, some osteoclasts formed on the surface of the graft with weak TRAP activity. In the HAP + PTH group, some osteoclasts formed on the surface of the graft and TRAP was activated.

The MT images were analyzed for tissue-specific area fractions (Figure 4a). The analysis revealed that, in the WH group, the amount of BG decreased, while the amount of NB increased as healing progressed. In the WH + PTH group, the amount of BG decreased the most at 4 weeks, and the formation of NB was minimal, whereas it was highest at 12 weeks as healing progressed. In the HAP group, there was no significant difference in the amount of BG and there was no significant difference in the formation of NB regardless of the healing period. In the HAP + PTH group, the amount of BG remained consistent regardless of the healing period, while the amount of NB gradually increased. At 4 weeks, the amount of NB was the lowest but, by 12 weeks, it surpassed that of the HAP group.

When comparing the amount of NB and BG, WH exhibited a gradual replacement of BG by NB as healing progressed (Figure 4b). WH + PTH demonstrated a similar ratio of NB to BG as WH at week 4, but more BG was replaced by NB as healing progressed through week 12. In contrast, in HAP, the ratio of NB to BG remained constant regardless of healing time. However, HAP + PTH displayed the highest ratio of BG at week 4, followed by some replacement of BG with NB as healing progressed to week 12.

## 4. Discussion

In this study, we compared bone regeneration using different grafts in an osteoporosis model and investigated the efficacy of directly applying PTH to the defect site. To accomplish this, we established a calvarial defect model in osteoporotic SD rats and assessed the effects of WH and HAP grafts, as well as PTH drug application, on the defect. The analysis showed a higher degree of regeneration in the WH graft than in the HAP graft within the range of error and a higher degree of regeneration than in the WH + PTH when PTH was applied.

Tissue area fraction analysis by tissue staining showed that WH, unlike HAP, tended to decrease in area as the remaining graft degraded toward week 12, while HAP showed little graft degradation (Figure 4a). In addition, in WH, NB formation occurred continuously and the value of NB + BG increased despite the decrease in BG, whereas, in HAP, the increase in NB was lower, resulting in a lower value of NB + BG compared to WH. This shows that the degradation of BG is more active and the formation of NB is more active in WH.

It can be seen that this degradation of BG is more evident with local application of the PTH drug. The area fraction of BG in the WH + PTH group demonstrates the lowest value at week 4 and continues to decline towards week 12. Consequently, the lowest BV/TV values observed during the initial 4 weeks in the WH + PTH group can be attributed to the substantial initial degradation of BG, resulting in a low BV (Figure 2b). As regeneration progresses in the WH + PTH group, an increasing amount of NB is formed, with week 12 exhibiting the lowest amount of BG and the highest amount of NB. This signifies the most active degradation of BG and the most vigorous formation of NB in the WH + PTH group. However, in the HAP group, even with PTH application, there is minimal degradation of BG, and the formation of NB at week 12 does not differ significantly from that in the HAP group.

This was also confirmed by TRAP staining (Figure 3c). TRAP staining showed that osteoclast activity was much higher in WH than in HAP at the beginning of regeneration at week 4, indicating active BG degradation. Furthermore, WH + PTH showed the highest level of osteoclast activity. TRAP expression is seen on the surface of the graft and, in the case of WH and WH + PTH, osteoclasts penetrate the graft and are activated. These TRAP staining results suggest that osteoclast-mediated degradation of the graft is active in WH and WH + PTH, with further enhancement in the presence of PTH.

The effect of osteoclast activation on the regeneration of the degraded graft is evident in the MT staining results at week 12 (Figure 3d). In both HAP and HAP + PTH groups, the boundary between the new bone and the graft is clearly defined, with new bone forming on the surface of the graft. However, in the case of WH, the boundary between the graft and the new bone is less distinct. New bone has penetrated and formed in areas where graft degradation has occurred, indicating active osteointegration. This phenomenon is more pronounced in WH + PTH group when PTH is applied. It is observed that graft material degradation is highly active with PTH application, leading to the replacement of graft material by new bone and subsequent bone integration.

Bone surface density data indicate that BS/TV increases as healing progresses in WH and WH + PTH groups but not in HAP and HAP + PTH groups (Figure 2c). This suggests that, in WH and WH + PTH groups, bone regeneration occurs throughout the defect area, leading to an increase in surface area. Conversely, in HAP and HAP + PTH groups, regeneration occurs primarily around the initially formed new bone and does not actively progress to new areas.

Furthermore, the bone surface density analysis reveals that BS/BV is higher in WH and WH + PTH than in HAP and HAP + PTH, indicating that the surface area of NB and BG is higher in WH (Figure 2d). This is likely attributed to the more active degradation of BG in WH, resulting in a higher surface area. In contrast, BG is less degraded in HAP, leading to a lower surface area.

These results suggest that WH graft degradation is more active than HAP graft degradation in the osteoporosis model, primarily due to early osteoclast activation during regeneration. Moreover, osteoclast activation influences new bone formation, with greater activation in the early stage leading to increased new bone formation in the later stages. These findings can be explained by osteoclast-induced resorption of the graft material, providing calcium and phosphorus ions to the bone regeneration site. Consequently, this normalizes the bone remodeling process by osteoblasts and osteoclasts, facilitating bone regeneration [22,23,24]. Additionally, the applied PTH, known to activate bone cells such as osteoclasts and osteoblasts, further accelerates this process [12,13,14,15]. Local treatment of PTH enhances osteoclast activation in the early regeneration stage, aiding in graft material resorption and subsequently increasing the degree of bone regeneration in the later stages of regeneration [13,25].

The progression of new bone regeneration from the surface of the graft to its interior was facilitated as the degradation of the graft was activated. This penetration of new bone into the graft was more pronounced with the application of PTH, which activated osteoclasts. Based on these results, it is evident that the application of bone graft in an osteoporosis model can lead to higher regeneration in the graft that can be more easily degraded by osteoclasts. Moreover, the direct application of PTH to the defect site can maximize new bone regeneration by stimulating osteoclasts in the early stages of regeneration and promoting osseointegration between the new bone and the graft. Consequently, this approach results in more efficient bone regeneration. Therefore, in the context of an osteoporosis model, implantation of a graft material such as WH, which degrades faster in the body compared to a slower-degrading material like HAP, offers advantages for regeneration. This advantage can be further augmented by local application of PTH.

## 5. Conclusions

The bone regeneration between WH and HAP grafts was analyzed under the OVX rat osteoporosis model. PTH(1-34), one of the anti-osteoporotic drugs, was directly applied to the defects to analyze bone regeneration in osteoporosis. In the osteoporosis model, bone regeneration was higher in the WH graft, which has a faster degradation rate in the body, than in the HAP graft. The application of PTH increased the degradation rate of the WH graft and increased bone regeneration. It is suggested that the activation of osteoclasts due to the degradation of bone graft material in the early stages of regeneration is beneficial for bone regeneration and osseointegration. Additionally, the application of PTH drugs can activate bone cells, including osteoclasts, resulting in a higher level of bone regeneration and osseointegration. In conclusion, for osteoporotic bone defects, bone graft materials that can be easily degraded by osteoclasts, such as WH, are more suitable. Additionally, local treatment with PTH can activate osteoclasts around the bone graft in the early stages of regeneration, leading to higher bone regeneration and improved osseointegration.

## Figures and Tables

**Figure 1 biomedicines-12-00820-f001:**
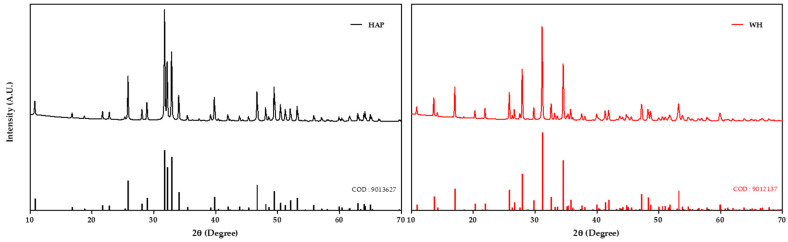
X-ray diffraction pattern data of HAP graft and WH graft. The measured data were matched with HAP (9013627) and WH (9012137) from Crystallography Open Database.

**Figure 2 biomedicines-12-00820-f002:**
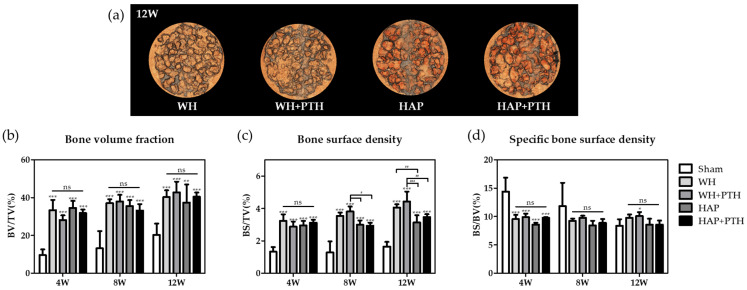
(**a**) Micro-CT reconstruction images of calvarial defects in 12 weeks, (**b**) bone and graft volume fraction (BV/TV), (**c**) bone and graft surface density (BS/TV), and (**d**) bone and graft specific surface density (BS/BV) (* *p* < 0.05 vs. Sham group, ** *p* < 0.01 vs. Sham group, *** *p* < 0.001 vs. Sham group, ^#^ *p* < 0.05, ^##^ *p* < 0.01, ^###^ *p* < 0.001, and ns is statistically non-significant).

**Figure 3 biomedicines-12-00820-f003:**
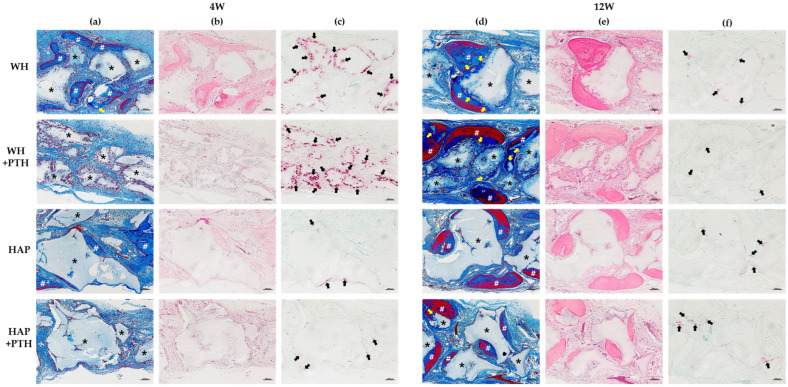
Histological analysis images of the defect area. (**a**) MT-stained images in 4 weeks, (**b**) H&E-stained images in 4 weeks, (**c**) TRAP-stained images in 4 weeks, (**d**) MT-stained images in 12 weeks, (**e**) H&E-stained images in 12 weeks, and (**f**) TRAP-stained images in 12 weeks (*: bone graft; #: new bone; yellow arrow: a site where graft material has been replaced by new bone; black arrow: osteoclast; scale bars: 200 μm).

**Figure 4 biomedicines-12-00820-f004:**
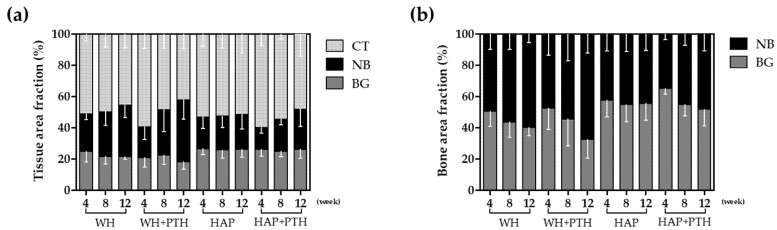
Histomorphometric analysis data from MT-stained images. (**a**) Tissue area fraction, (**b**) bone area fraction (CT: connective tissue; NB: new bone; BG: remaining bone graft).

## Data Availability

Data are available from the authors upon request.

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
