# Peer review of "The Effects of Local Treatment of PTH(1-34) and Whitlockite and Hydroxyapatite Graft to the Calvarial Defect in a Rat Osteoporosis Model"

_biomedicines, 2024, doi:10.3390/biomedicines12040820_

Round 1
Reviewer 1 Report
Comments and Suggestions for Authors
The manuscript is interesting but some points need to be clarified. Do you have any evidence of the long-term effects of your treatment on rats?
Introduction: Line 39 please integrate with the note related to the novel mechanism of actions of bisphosphonate mediated by inhibition of KATP channels in the skeletal muscle fibers and bone cells responsible for the musculoskeletal ADR (Scala R et al., 2022) and the novel mechanism of action mediated by the modulation of TRPV1 channel in osteoblast (Scala R et al., 2019).
Zoledronic Acid as a Novel Dual Blocker of KIR6.1/2-SUR2 Subunits of ATP-Sensitive K(+) Channels: Role in the Adverse Drug Reactions.
Maqoud F, Scala R, Tragni V, Pierri CL, Perrone MG, Scilimati A, Tricarico D. Pharmaceutics. 2021 Aug 27;13(9):1350. doi: 10.3390/pharmaceutics13091350.
Zoledronic Acid Modulation of TRPV1 Channel Currents in Osteoblast Cell Line and Native Rat and Mouse Bone Marrow-Derived Osteoblasts: Cell Proliferation and Mineralization Effect.
Scala R, Maqoud F, Angelelli M, Latorre R, Perrone MG, Scilimati A, Tricarico D. Cancers (Basel). 2019 Feb 11;11(2):206. doi: 10.3390/cancers11020206.
Line 136 Method
Some additional statistical analysis is needed to support the conclusion of this work
you need to calculate the power of your study by power analysis
Figure 1 needs to be revised, traces 2 and 4 seem off scale
Figure 3 The symbol @ is not conventionally used
Line 325 Please replace one of the anti-osteoporotic drugs
Author Response
Thank you for your time and consideration on our submission. We have answered your comments. The details follow your original comments.
Comments1: The manuscript is interesting but some points need to be clarified. Do you have any evidence of the long-term effects of your treatment on rats?
Response1: Thank you for your comment. The treatment method of applying the drug used in this study, PTH, directly to the bone defect site, is not extensively researched, so it cannot be conclusively stated about the long-term effects beyond the 12-week results shown in this paper. However, PTH(1-34) has been reported to have effects on alleviating osteoporosis and promoting bone recovery in SD rats with long-term use (KIMMEL, Donald B., et al. Endocrinology, 1993, 132.4: 1577-1584.). Based on the long-term effects of PTH(1-34) and the results of this study, it is predicted that even with observations extending beyond 12 weeks, the WH+PTH group will show the highest level of bone recovery, as evidenced by the highest levels of bone union and regeneration in that group at the 12-week mark.
Comments2: Introduction: Line 39 please integrate with the note related to the novel mechanism of actions of bisphosphonate mediated by inhibition of KATP channels in the skeletal muscle fibers and bone cells responsible for the musculoskeletal ADR (Scala R et al., 2022) and the novel mechanism of action mediated by the modulation of TRPV1 channel in osteoblast (Scala R et al., 2019).
Response2: Thank you for your suggestions. We have added the reference paper (Scala et al., 2022) and the general action and use of bisphosphonate drugs.
Comments3: Line 136 Method. Some additional statistical analysis is needed to support the conclusion of this work. you need to calculate the power of your study by power analysis.
Response3: Thank you for pointing this. We conducted a post-hoc power analysis, and the results showed that all statistical analyses had a power exceeding 0.8. Therefore, we added the following statement: "Post-hoc power analysis using G*Power software showed that all statistical analyses had a power exceeding 0.8."
Comments4: Figure 1 needs to be revised, traces 2 and 4 seem off scale.
Response4: Thank you for pointing this. We adjusted and improved the scale of Figure 1 to make it more visually accessible.
Comments5: Figure 3 The symbol @ is not conventionally used.
Response5: Thank you for pointing this. We changed the symbol to #.
Comments6: Line 325 Please replace one of the anti-osteoporotic drugs
Response6: Thank you for pointing this. We replaced an incorrect sentence.

Reviewer 2 Report
Comments and Suggestions for Authors
The text is not always clear. For instance: ''The effect of osteoclasts was more activated, and bone regeneration was enhanced 21 PTH was applied'' what does it mean?
In the abstract you should explain which analyses you have carried out.
What does this sentence mean ''. Osteoporosis, in particular, has a prevalence rate of over 18% worldwide across all age groups and is steadily increasing every year''?
What DS rats stand for?
For the bone grafting preparation, was there a specific ratio between the volume of liquid and the one of the sponge?
Was the PTH drug in liquid form? How did you ensure that the drug was absorbed throughout the graft if the graft was already put in the defect?
Line 117, please specify which formalin did you use for storage? Manufacturer? Concentrations?
Describe the procedure used for decalcification of the tissues.
Paragraph 2.5 check the capital letters.
Could you include SEM images of the scaffolds?
Why the following analysis was carried out? BS/TV and BS/BV? Please explain the importance in the results section.
Explain what we can visualize with the different dyes. What about highlightening the presence of osteoclasts in the photos?
Comments on the Quality of English LanguageThe English must be revised in the manuscript.
Author Response
Thank you for your time and consideration on our submission. We have answered your comments. The details follow your original comments.
Comments1: The text is not always clear. For instance: ''The effect of osteoclasts was more activated, and bone regeneration was enhanced 21 PTH was applied'' what does it mean?
Response1: Thank you for your point this. We revised the abstract section.
Comments2: In the abstract you should explain which analyses you have carried out.
Response2: Thank you for your point this. We revised the abstract section.
Comments3: What does this sentence mean ''. Osteoporosis, in particular, has a prevalence rate of over 18% worldwide across all age groups and is steadily increasing every year''?
Response3: Thank you for your point this. We revised the sentence to “Osteoporosis, in particular, has a global prevalence rate exceeding 18% across all age groups, with a steady annual increase.”.
Comments4: What DS rats stand for?
Response4: Thank you for your comment. We changed this to “SD (Sprague Dawley) rats”.
Comments5: For the bone grafting preparation, was there a specific ratio between the volume of liquid and the one of the sponge?
Response5: Thank you for your point this. To ensure even coating of the sponge with the slurry, it is fully immersed in the slurry, enabling maximum absorption. Typically, a sponge with a volume of 15 ml absorbs approximately 10 ml of slurry. Section 2.1 has been updated to include the phrase "fully immersed in".
Comments6: Was the PTH drug in liquid form? How did you ensure that the drug was absorbed throughout the graft if the graft was already put in the defect?
Response6: Thank you for your point this. PTH was dissolved in saline and applied at a volume of 20μl per 30mg of graft. The graft used had a wettability of more than 100% liquid absorption per weight, which ensured that all drug was absorbed by the graft and did not spill over. We have added this information to Section 2.3 and added the wettability data to Supplementary Material as Figure S2.
Comments7: Line 117, please specify which formalin did you use for storage? Manufacturer? Concentrations?
Response7: Thank you for your point this. We added the information in Section 2.3.
Comments8: Describe the procedure used for decalcification of the tissues.
Response8: Thank you for your point this. We added the information in Section 2.5.
Comments9: Paragraph 2.5 check the capital letters.
Response9: Thank you for your point this. We revised the section 2.5.
Comments10: Could you include SEM images of the scaffolds?
Response10: Thank you for your comment. SEM images of the bone graft are included in the Supplementary Material as Figure S1.
Comments11: Why the following analysis was carried out? BS/TV and BS/BV? Please explain the importance in the results section.
Response11: Thank you for your comment. We analyzed the pattern of defect regeneration using BS/TV. BS/TV provides information on the surface of bone formed within the defect volume, indicating whether bone formation occurs at specific sites or is widespread across multiple sites. We also analyzed the characteristics of the regenerated bone using BS/BV. BS/BV informs about the density of bone formed, indicating whether the regenerated bone has a high density or is structured with a high surface area. We added the explain in section 3.3.
Comments12: Explain what we can visualize with the different dyes. What about highlightening the presence of osteoclasts in the photos?
Response12: Thank you for your comment. We added a description of the staining method to Section 2.5. We also marked osteoclasts with a black arrow in Figure 3.
Comments: The English must be revised in the manuscript.
Response: Thank you for point this. We have clearly revised the manuscript text.

Round 2
Reviewer 2 Report
Comments and Suggestions for Authors
The manuscript was greatly improved
Comments on the Quality of English LanguageNo issue